# Direct observation of single-molecule hydrogen-bond dynamics with single-bond resolution

Ce Zhou[1], Xingxing Li[2], Zhongliang Gong[3], Chuancheng Jia[1], Yuanwei Lin[1], Chunhui Gu[1], Gen He[1], Yuwu Zhong[3], Jinlong Yang[2] & Xuefeng Guo[1,4]

The hydrogen bond represents a fundamental interaction widely existing in nature, which plays a key role in chemical, physical and biochemical processes. However, hydrogen bond dynamics at the molecular level are extremely difficult to directly investigate. Here, in this work we address direct electrical measurements of hydrogen bond dynamics at the single-molecule and single-event level on the basis of the platform of molecular nanocircuits, where a quadrupolar hydrogen bonding system is covalently incorporated into graphene point contacts to build stable supramolecule-assembled single-molecule junctions. The dynamics of individual hydrogen bonds in different solvents at different temperatures are studied in combination with density functional theory. Both experimental and theoretical results consistently show a multimodal distribution, stemming from the stochastic rearrangement of the hydrogen bond structure mainly through intermolecular proton transfer and lactam–lactim tautomerism. This work demonstrates an approach of probing hydrogen bond dynamics with single-bond resolution, making an important contribution to broad fields beyond supramolecular chemistry.

[1] Beijing National Laboratory for Molecular Sciences, State Key Laboratory for Structural Chemistry of Unstable and Stable Species, College of Chemistry and Molecular Engineering, Peking University, Beijing 100871, China. [2] Hefei National Laboratory for Physics Sciences at the Microscale, University of Science and Technology of China, Hefei, Anhui 230026, China. [3] Beijing National Laboratory for Molecular Sciences, CAS Key Laboratory of Photochemistry, CAS Research/Education Center for Excellence in Molecular Sciences, Institute of Chemistry, Chinese Academy of Sciences, Beijing 100190, China. [4] Department of Materials Science and Engineering, College of Engineering, Peking University, Beijing 100871, China. Ce Zhou, Xingxing Li and Zhongliang Gong contributed equally to this work. Correspondence and requests for materials should be addressed to Y.Z. (email: zhongyuwu@iccas.ac.cn) or to J.Y. (email: jlyang@ustc.edu.cn) or to X.G. (email: guoxf@pku.edu.cn)

The dynamics of hydrogen bonds, which are ubiquitous in nature from elegant base-pair interactions in DNAs to sophisticated supramolecular assemblies, represents an ever-present question associated with the processes of chemical reactions and biological activities[1]. Many studies have been carried out by using spectroscopic methods and hydrogen/deuterium substitution where the signals stem from the ensemble experiments and the hydrogen bond dynamics were deduced indirectly[2–4]. Scanning probe microscopy (SPM) is suitable to study hydrogen bond molecules on the surface as it provides submolecular resolution and also allows controllable manipulation of single molecules by various stimuli. Remarkably, scanning tunnelling microscopy (STM) and dynamic force microscopy (DFM) have been used to provide molecular resolution imaging, thus enabling the visualisation of the chemical structures of hydrogen bonds[5–8]. Intramolecular hydrogen-transfer reactions and related tip-induced reactions have been studied on the surface in ultrahigh vacuum conditions by using low-temperature STM[9–13]. However, until now these dynamic processes still remain poorly understood at the single-molecule level because hydrogen bond dynamics under normal conditions, such as bond rearrangements and hydrogen/proton transfer processes, are extremely difficult to directly probe in solvents. There are only a few examples where the sensitivity of electron tunnelling through hydrogen bonds to the perturbations from the measurement environments, such as solvent, temperature, the measurement distance, external electric field or molecular conformation and tautomerisation, was utilised to develop functional electrical devices with superior sensitivity and recognition capability[14–16], reflecting the importance of building intrinsic models of hydrogen bond dynamics from the single-molecule insight.

To directly uncover heterogeneous molecular behaviours in chemical and biological reactions that are usually inaccessible in ensemble experiments, several approaches have been developed to realise single-molecule electrical measurements of molecular interactions by using different device architectures, such as nanotubes[17–20], nanowires[21,22], nanopores[23] and STM break junctions[14]. Among these approaches, single-molecule techniques[24,25], in particular graphene electrode-molecule single-molecule junctions (GM-SMJs)[26], are promising because they are able to covalently integrate individual molecular systems tested as the conductive channel into electrical nanocircuits, thus solving the key issues of the device fabrication difficulty and the poor stability. These techniques prove to be a robust platform of single-molecule electrical detection that is capable of probing the dynamic processes of chemical reactions at the single-event level with high temporal resolution and high signal-to-noise ratios, for example photoinduced conformational transition[27], temperature-dependent σ-bond rotation[28] and host–guest interaction[29]. In this work, for the first time we represent direct real-time monitoring of single-molecule hydrogen bond dynamics on the basis of the platform of GM-SMJs, where a quadruple hydrogen bond supramolecular system is covalently sandwiched between graphene point contacts to form stable supramolecule-assembled SMJs.

## Results

**Device fabrication and electrical characterisation.** Single-layered graphene was synthesised via a chemical vapour deposition process on copper foils. After graphene films were transferred to SiO$_2$/Si wafers, metal electrodes were subsequently patterned by using photolithography. The nanogapped graphene point contact arrays with carboxylic acid groups on each side were fabricated by a dash-line lithographic (DLL) method[26]. In order to investigate the hydrogen bond dynamic process with

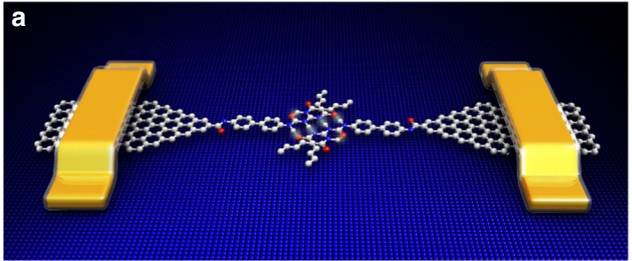

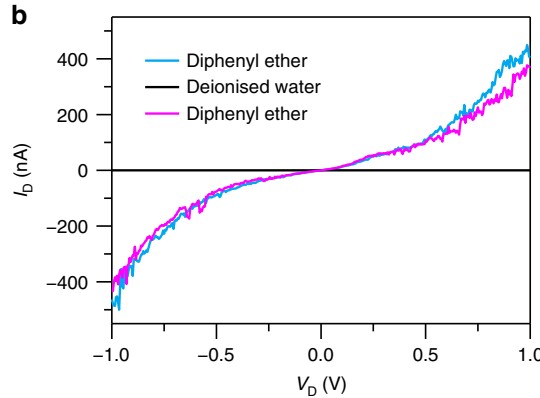

**Fig. 1** Device structure and electrical characterisation of HBB-SMJs. **a** Schematic representation of HBB-SMJs. **b** I–V curves of a HBB-SMJ sequentially immersed in diphenyl ether (blue), deionised water (black) and diphenyl ether again (magenta). $I_D$ source-drain current, $V_D$ source-drain voltage

solvent and temperature dependence, we constructed stable hydrogen bond-bridged SMJs (HBB-SMJs) according to the previous self-assembly method (Fig. 1a)[26]. A quadruple hydrogen bond dimer based on ureido pyrimidine-dione (UPy) was chosen because of two following reasons: One is that a UPy dimer should behave as a good conducting channel by forming a strong donor–donor–acceptor–acceptor array of intermolecular quadruple hydrogen bonding[30], which is a prerequisite to realise long-term real-time monitoring of the hydrogen bond dynamic behaviour; The other is due to the degenerate prototropy effect, only a set of single duplexes can be formed, simplifying the system we studied[31]. Each monomer can covalently link the amino substituent on UPy with carboxylic groups on the edges of graphene point contacts by employing a 1-ethyl-3-(3-dimethylaminopropyl) carbodiimide (EDCI) coupling protocol in pyridine, respectively. More details of molecular synthesis and the device fabrication process are provided in Supplementary Note 1 and Supplementary Figs. 1−3. Because in general they dimerise in weak polar solvents such as diphenyl ether and dissociate in strong polar solvents such as water, alternate diphenyl ether and water treatments enable the on–off switching of the conductance of HBB-SMJs. As demonstrated in Fig. 1b and Supplementary Fig. 4, the successful establishment of HBB-SMJs was identified by the resurgence of the electrical current when sequentially treating the devices in diphenyl ether and water. The optimised connection yield was found to be ~18%. Note that statistical analysis used in a previous work confirmed that the ratio of single-junction devices to the overall reconnected devices is over 90% and electron transfer is through hydrogen bonds between single molecules (also see the Supplementary Note 2).

**Real-time electrical measurement.** Before real-time testing, each device was in turn measured in diphenyl ether and water to ensure that the current signal originated from hydrogen bonding.

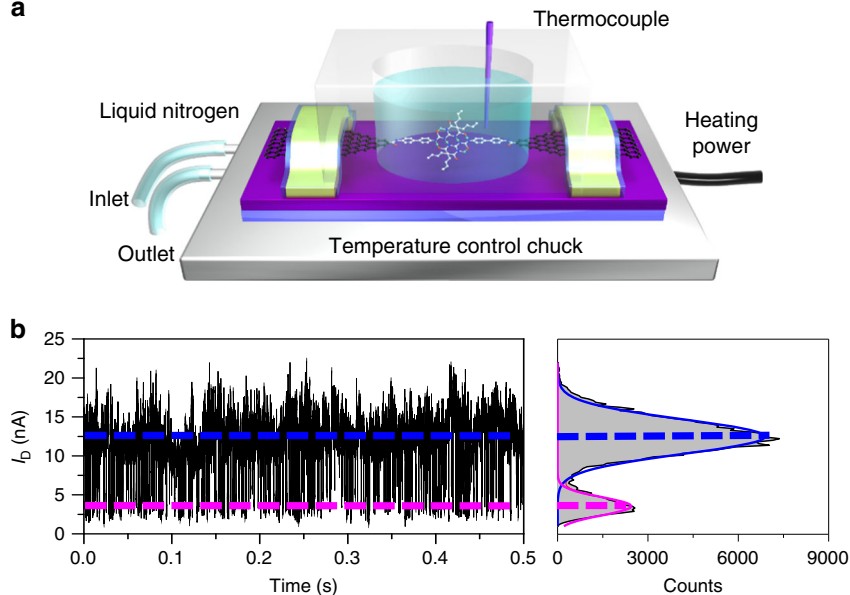

**Fig. 2** Measurement setup and real-time current recordings of hydrogen bond dynamics. **a** Schematic illustration of the electrical measurement setup. **b** Real-time current recordings of hydrogen bond dynamics during 500 ms measured in TeCA at 273 K. The right panel is the corresponding histogram of current values, showing a bimodal current distribution. $V_{bias} = 300$ mV

With a temperature controlling module and a poly-dimethylsiloxane (PDMS) solvent reservoir (Fig. 2a), time-dependent electrical characterisation of HBB-SMJs was carried out at the solid–liquid interface while HBB-SMJs were immersed in solvents at a certain temperature. All the real-time electrical measurements were conducted at a source-drain bias ($V_{bias}$) of 300 mV and a zero gate bias with a sampling rate of 57.6 kHz. We found that the obtained current-time ($I$–$t$) curves show a series of regular large-amplitude current fluctuations with strong temperature and solvent dependence. As shown in Fig. 2b, both the $I$–$t$ curve and the corresponding current-count histogram in 1,1,2,2-tetrachloroethane (TeCA) at 293 K reveal a typical bimodal distribution centred at ~12.5 nA and ~3.4 nA, respectively.

**Solvent-dependent and temperature-dependent measurements**. Since solvent and temperature play the important roles in complicated hydrogen bond interactions, to clarify the dynamic mechanism, we focused the studies on the experiments in two solvents with three different temperatures. As shown in Fig. 3, current signal features highly depend on different solvents. Two large-amplitude peaks were detected in TeCA while a multimodal distribution (at least four obvious peaks) was observed in diphenyl ether, suggesting different mechanisms of intermolecular interactions through hydrogen bonding in different solvents (as discussed in detail below).

We observed similar fluctuation behaviours in the tests of at least five devices. Supplementary Figs. 5−7 show another data set, demonstrating the reproducibility. Note that previous studies reported the observation of similar current fluctuations in electronic devices, in particular carbon nanotube devices in which localised trap states interact with conducting channels, again proving the reliability[32–34]. However, unlike these devices based on carbon nanotubes or nanowires, in the current case the device conductance is dominated by an UPy-based supramolecular system and the trap states existing in defects of graphene electrodes or the interface between graphene and SiO$_2$ have a negligible influence on the conductance changes. To support this

fact, careful control experiments by using partially cleaved graphene ribbon devices were carried out under the same experimental condition. These control experiments demonstrated that the $I$–$t$ data did not exhibit particular fluctuations (Supplementary Fig. 8), thus excluding the possibility of trapping effects in the system. To rule out the possibility that the signals originate from stochastic bond fluctuations, a control molecule based on a sexiphenyl structure, which has similar conjugation/conductance values and the same amino end groups, was designed and synthesised. No particular fluctuations were observed in control devices reconnected by this molecule under the same experimental condition (Supplementary Fig. 9). Furthermore, similar on–off switching by different solvent treatments and two-level fluctuations in $I$–$t$ measurements were also observed in HBB-SMJs based on a double-hydrogen bond supramolecular system (Supplementary Fig. 10). Collectively, these results confirmed that the observed stochastic switching fluctuations in HBB-SMJs should originate from the supramolecular system and its possible interactions with the surrounding liquid environment and the electric field applied.

**Statistical and theoretical analyses in TeCA**. In general, the $I$–$t$ curve shows a rapid, large-amplitude two-state fluctuation in TeCA at 273 K. The resulting current-count histograms (Fig. 3a) reveal a bimodal distribution. The conductance of the 'low' state is much smaller than that of the 'high' state, indicating a weaker hydrogen bonding system. By using a hidden Markov model, we extracted the distribution of dwell times in the high-conductance and low-conductance states and the lifetimes from the dwell-time histogram (Supplementary Fig. 11). These plots can be fitted with a single-exponential function, with average lifetimes of $\tau_{high} =$ ~0.660 ± 0.042 ms and $\tau_{low} =$ ~0.581 ± 0.042 ms for the high-conductance and low-conductance states, respectively.

In order to investigate the evolution of the current fluctuations in the HBB-SMJ system, the $I$–$t$ curves of HBB-SMJs immersed in TeCA were recorded at three different temperatures. Similar current spikes at the submillisecond level and bimodal distributions were observed at each temperature (Fig. 3a, c). These

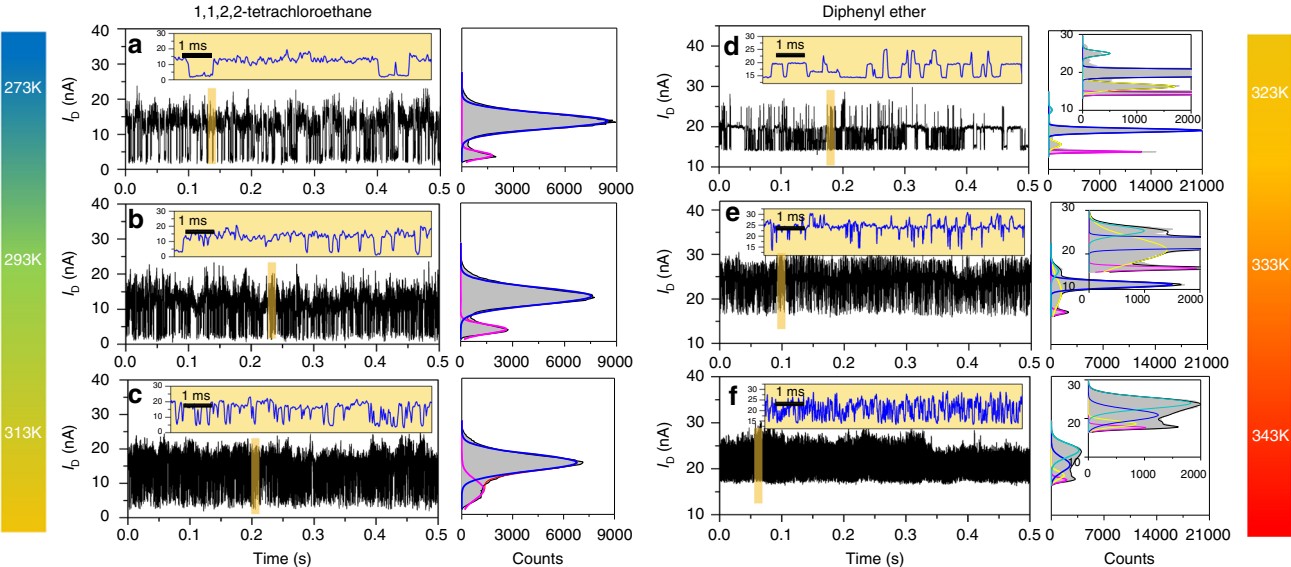

**Fig. 3** Temperature-dependent and solvent-dependent measurements of hydrogen bond dynamics. *I–t* curves and the corresponding histograms of a HBB-SMJ device immersed in TeCA at 273 K (**a**), 293 K (**b**) and 313 K (**c**), and in diphenyl ether at 323 K (**d**), 333 K (**e**) and 343 K (**f**). Insets in *I–t* curves show the corresponding enlarged parts marked in yellow. Insets in the histograms amplify the details of the current distributions. $V_{bias} = 300$ mV

idealised two-level fluctuations were analysed to provide a set of lifetimes ($\tau_{high}$ and $\tau_{low}$) and the corresponding interconversion rate constants ($k_1 = 1/\tau_{low}$ and $k_2 = 1/\tau_{high}$) at different temperatures (Supplementary Table 1). Obviously, the current oscillation has the strong temperature dependence. Both rate constants increase progressively with increasing the temperature. When the temperature increases, the current-count distributions between the 'high' and 'low' states change gradually and the areas of the 'low' state increase.

It is unreasonable to directly attribute the current fluctuations to the association and dissociation processes of the UPy dimer because of the following three reasons: First, the low state still has a conductivity at a nA scale in TeCA while the dimer completely dissociates in water with no conductivity (<1 pA) as shown in Fig. 1b. Second, the width of the Gauss distribution of the low state illustrates a level of heterogeneity, which indicates that the low state is not a simple dissociated state. At last, the significant downfield NH chemical shifts (12.58, 10.68 and 9.86 ppm) were observed in 1,1,2,2-Tetrachloroethane-$d_2$ at 313 K (Supplementary Note 1), indicating that there were strong hydrogen bonding interactions[23]. The UPy-based quadruple hydrogen bonding dimers always have high stability in the low-polar solvent. The dimerisation constant of similar AADD-type UPys in CHCl$_3$ can reach $10^4$ M$^{-1}$ to $10^7$ M$^{-1}$ [23,24]. We also built a simple theoretical model to qualitatively illustrate the difficulty of the dissociation processes by solvent intercalation. The intercalation energy of TeCA was calculated to be ~0.81 eV, indicating that the dissociation processes hardly exist under our experimental condition (Supplementary Fig. 14). On the basis of experimental and theoretical simulation results, an intermolecular proton transfer mechanism is preliminarily proposed, which will be discussed in detail later.

**Statistical and theoretical analyses in diphenyl ether**. We next turn our attention to reveal the dynamic process in diphenyl ether. In general, a sharp decay of the hydrogen bonding conductance indicates a longer bond length. However, we found that the conductance changes within 50% and the lowest state maintains a good conductance, suggesting that the quadrupolar

hydrogen bonding structure should maintain during real-time measurements. The intercalation energy of diphenyl ether was calculated to be ~0.94 eV at 323 K, revealing that the diphenyl ether is much harder to intercalate into the dimer than TeCA. In addition, the intercalation energy of diphenyl ether is much higher than the intermolecular proton transfer barrier (~0.35 eV) (Supplementary Figs. 14 and 16). These calculations demonstrate that the HBB-SMJ in diphenyl ether prefers to take a proton transfer process rather than dissociation, which is in agreement with the experiments (as detailed below).

It is remarkable that different current spikes and stages were observed at each temperature (Fig. 3d, f). We obtained the multiple levels of the current signals with different distributions at 323 K and 333 K, respectively. When the temperature reached 343 K, the fluctuation rate approached the temporal resolution of the instrument. At 323 K, we achieved a clear current distribution, indicating that several microstates of HBB-SMJs can be distinguished in this condition. These microstates should provide a large amount of information for us to study the dynamic process. To illustrate the uniform fluctuation characteristics in diphenyl ether at 323 K, the conductance data are presented at four different time-scale magnifications (Fig. 4). In this way, the spikes and corresponding plateaus at different amplitudes can be clearly presented. For the convenience of further analysis, we classified the spikes and plateaus as four microstates by amplitude as Fig. 4d depicts.

## Discussion

The current signals, labelled as States 1–3, showed stage-like oscillation characteristics, which can last as long as several milliseconds. These stages should be assigned to some metastable structures while maintaining the quadrupolar hydrogen bonding structure. It is well known that for molecular junctions, regular changes in conductance could take place due to their specific responses to the changes of molecular conformation/orientation[21,35], chemical reactions[19,36] or tautomerisation[37,38]. Under the experimental conditions in the current case, two kinds of possible processes could occur and result in several metastable structures without damaging the quadruple hydrogen bond array.

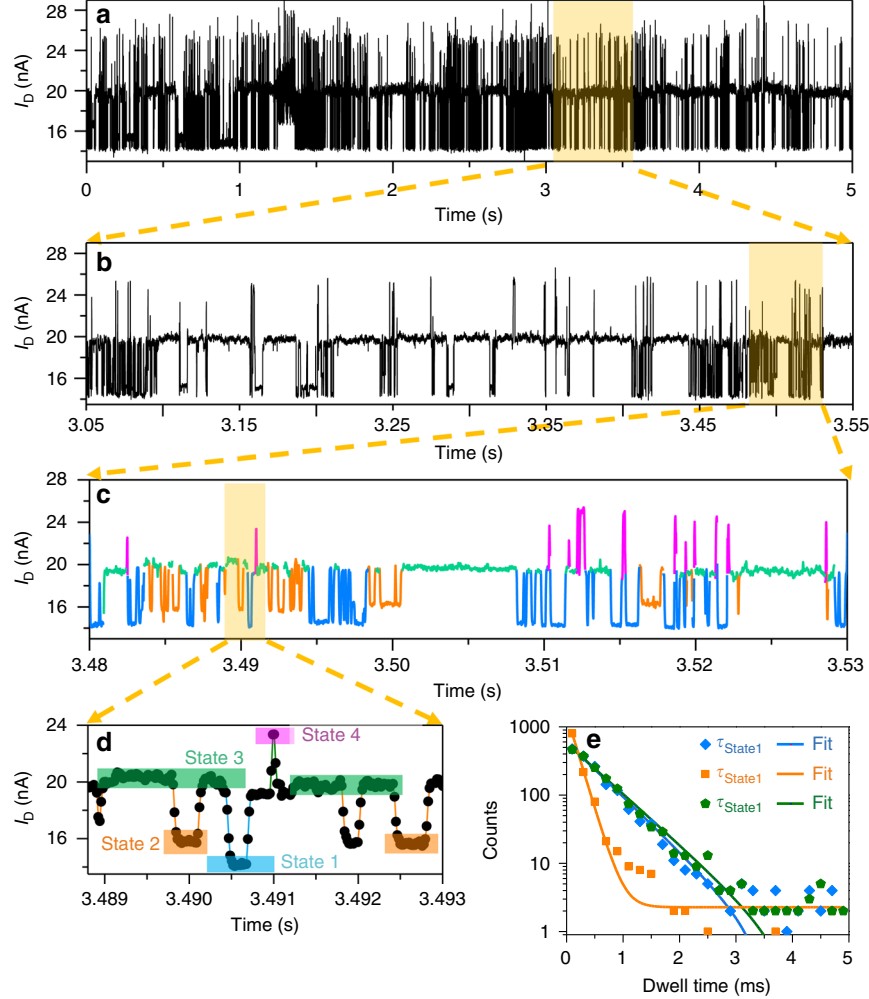

**Fig. 4** Kinetic analyses of hydrogen bond dynamics in diphenyl ether. *I*–*t* curves of a HBB-SMJ device in diphenyl ether at 323 K at four different time scales (**a**–**d**) and the corresponding dwell-time histogram of States 1–3 in (**e**), showing a single-exponential decay fitting with different lifetimes. $V_{bias} = 300$ mV

One possibility that may take place in the junction is intermolecular proton transfer among the terminal electronegative O/N atoms in quadruple hydrogen bonds. Such an intermolecular proton transfer process results in different proton distributions between the two binding monomers, which in turn significantly changes the electron transport property of the junction. The other is lactam–lactim tautomerism, in which the carbonyl group ($-C=O$) in the dimer undergoes a tautomerisation to become an enol group ($=C-OH$). Such a tautomerisation process is commonly observed in similar heterocyclic ring systems[39–41]. The lactam and its lactim tautomer have very different conjugations, which could in principle lead to the different electron transport properties. We also extracted the distribution of dwell times of States 1–3 and the lifetimes from the dwell-time histogram (Supplementary Fig. 12). These plots can be fitted with a single-exponential function (Fig. 4e), providing total average lifetimes of $\tau_{State\ 1} = {\sim}0.545 \pm 0.022$ ms, $\tau_{State\ 2} = {\sim}0.194 \pm 0.022$ ms and $\tau_{State\ 3} = {\sim}0.585 \pm 0.018$ ms for each state with minor errors, respectively. These values are consistent with previous works[40,42], which, however, actually reported the broad range from seconds to nanoseconds, highly depending on the measurement conditions, such as the object of study, the method used, the solvent, the temperature and so on. The corresponding interconversion

rate constants between different states at different temperatures are listed in Supplementary Table 2. Similarly, the current oscillation had a strong temperature dependence and the switching frequency of the states increased progressively with increasing the temperature.

Unlike current spikes of States 1–3, State 4 showed spike-like, high-frequency features with an extremely short lifetime and the amplitude concealed a wide, nonstationary distribution, which broadened as the temperature increased. These characteristics imply that the quadrupolar hydrogen bond system had been instantaneously perturbed, which was commonly observed in single-molecule/single-event detection[32]. Although the detailed mechanism of this behaviour cannot be decided yet, some explanations can be reached from our observations and analysis. We interpret that the surrounding diphenyl ether molecules could induce the alteration of the hydrogen bonding structure, thus resulting in dynamic conductance changes of SMJs. In other words, the solvent plays an important role in the conformation of individual hydrogen-bonded supramolecules through several interactions such as π–π stacking, hydrogen bonding interaction and dipole–dipole interaction. It is preliminarily deduced that the stabilising effects, caused by diphenyl ether molecules that are bound to the junctions and generally weaken hydrogen bonding,

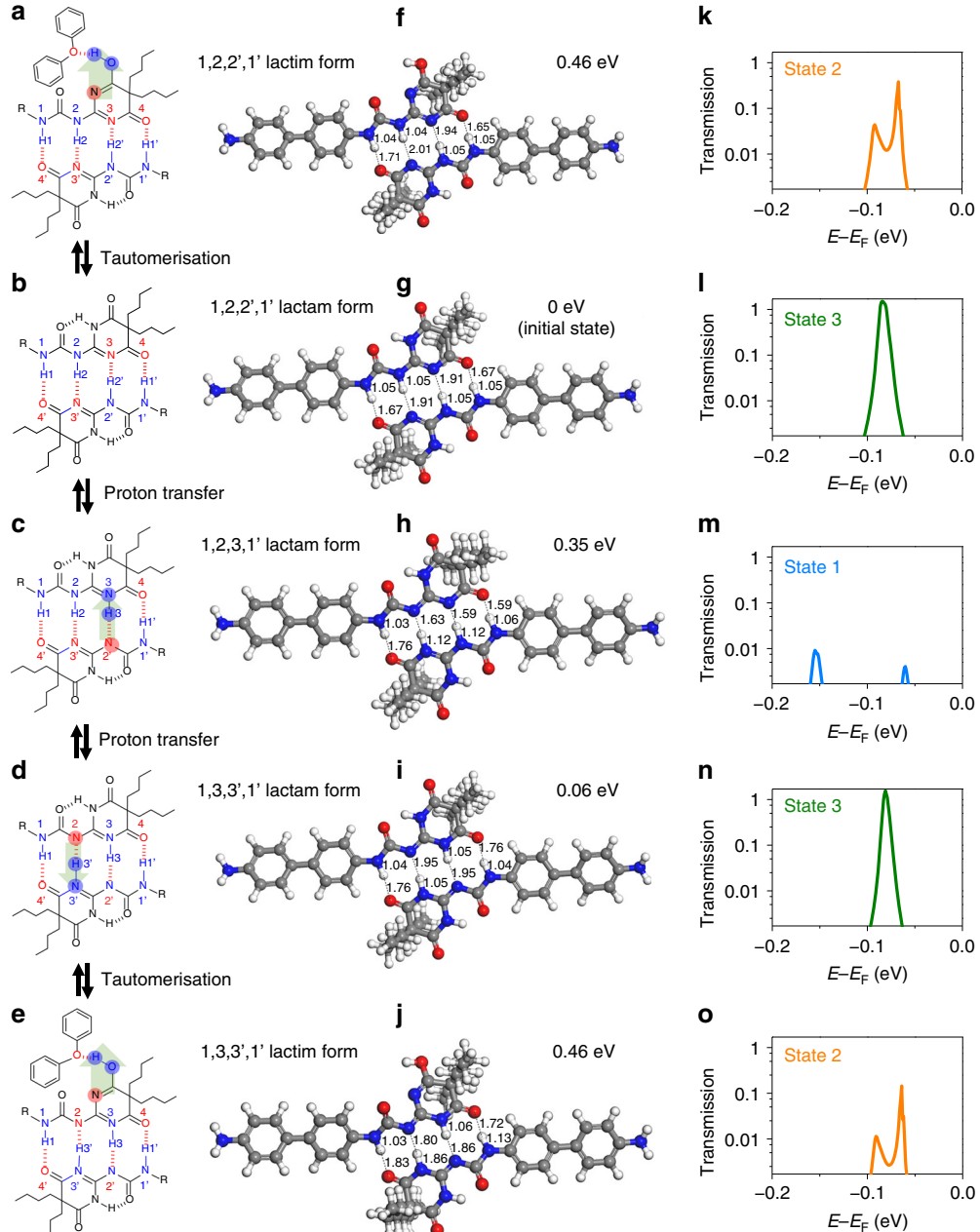

**Fig. 5** Theoretical model of hydrogen bond transformations. Schematic diagram of each transformation process (**a–e**) with five low-lying energy structures (**f–j**) and corresponding transmission spectra at a zero bias voltage (**k–o**)

might transiently work when the surrounding solvent environment is altered by an electric field and physical perturbations, thus finally leading to a higher conductance.

To determine the metastable structures, we have carried out systematic theoretical studies for possible configurations by either an intermolecular proton transfer reaction or tautomerism (Supplementary Fig. 15). After substantial theoretical calculations, five low-lying energy structures, including the most stable initial state (referred as 1,2,2',1' lactam form), were identified (Fig. 5). In the first kind of the intermolecular proton transfer process, there are two metastable structures, 1,3,3',1' lactam form and 1,2,3,1' lactam form, which are located just 0.06 and 0.35 eV above the initial state, respectively. Under lactam–lactim tautomerism, these three structures can be converted into their corresponding lactim forms, among which two low-energy lactim structures with the

energy only 0.46 eV above the initial state, named 1,2,2',1' and 1,3,3',1' lactim forms, have been determined. Note that the diphenyl ether solvent plays an important role in stabilising the lactim forms by forming a new hydrogen bond between the hydroxyl of the lactim forms and the oxygen atom of the diphenyl ether. This can be readily seen from the test calculation without considering any solvent, which gives a much higher energy of 0.94 eV for the 1,2,2',1' lactim form.

To further explore the feasibility of interconversions among the five structures, we have calculated the lowest energy reaction pathways (Supplementary Fig. 16), and found that for the first kind of the intermolecular proton transfer process, the reaction barriers are rather small (~0.35 eV), which indicates that proton can readily migrate between the terminal electronegative O/N atoms in the quadruple hydrogen bond at room temperature.

While for lactam–lactim tautomerism, a somewhat higher energy barrier of about 0.67 eV was obtained, which can also be overcome at room temperature. Moreover, previous studies of similar hydrogen bonding dimers have shown that a local electric field (bias voltage) can increase the rate constant of the tautomeric equilibrium by lowering the barrier height[37,43,44]. Therefore, it is expected that these five structures can emerge alternately in response to thermal and electric stimuli under our experimental conditions.

In order to gain better understanding of the correlation between the current stages and configurations, the transmission spectra at a zero bias voltage were calculated for all the five structures (Fig. 5; Supplementary Fig. 17). The transmission spectra of some configurations were found to be significantly different near the Fermi level, leading to different conductance and current stages. In combination with the calculated conductance and the relative energies with the statistical results from experiments, we were able to assign each current stage to each configuration (as shown in Fig. 4) and build the whole transformation process.

Figure 5 demonstrates the whole transformation process. In the first step, from the initial state (State 3), 1,2,2',1' lactam form, the hydrogen atom on N2' is reversibly transferred to N3, forming a dissymmetric tautomer, 1,2,3,1' lactam form, with low conductivity (State 1). In the second step, a symmetric tautomer, 1,3,3',1' lactam form with high conductivity and stability, is formed when the hydrogen atom on N2 reversibly migrates to N3'. However, in the electrical signal, it is difficult to distinguish the two symmetric structures, i.e. 1,2,2',1' and 1,3,3',1' lactam forms, due to their very similar conductance. Even though we could occasionally detect the difference between the two symmetric structures as shown in Fig. 4d, in most cases, they share the same conducting state of State 3. Similarly, under lactam–lactim tautomerism, two lactim forms of 1,2,2',1' and 1,3,3'1' structures, are rarely differentiated in experiments. They constitute the intermediate conducting state of State 2. Note that the lactim forms are much more unstable than the corresponding lactam forms due to the breakage of intramolecular hydrogen bond in the lactam–lactim tautomerism process. As pointed out above, they lie about 0.46 and 0.11 eV higher in energy than 1,2,2',1' and 1,2,3,1' lactam forms, respectively, consistent with the experiment results that the mean duration of State 2 is much shorter than State 3 and State 1. Occasionally, we also found that State 2 cannot be detected in the same experimental conditions (Supplementary Fig. 5), indicating that the lactam–lactim tautomerism process occurs nearly simultaneously on the timescale of the electronic measurements and the process is highly related to the configurations of HBB-SMJs and their surrounding diphenyl ether molecules.

However, the lactam–lactim tautomerism processes were hard to be observed in TeCA possibly because TeCA cannot stabilise the lactim form by forming hydrogen bonds like diphenyl ether. Theoretical calculation results showed that the energy of the lactim form in TeCA was ~1.11 eV above the basic state and the reaction barrier was also very high (~1.26 eV) (Supplementary Fig. 18). This well explains the reason why we only observed the two-state fluctuations in TeCA.

This work demonstrates the capability of intrinsically transducing the exquisite hydrogen bond dynamic process into real-time electrical signals. Our systematic experimental and theoretical studies consistently show and interpret stochastic transitions of a classical quadrupolar hydrogen bonding system through intermolecular proton transfer and lactam–lactim tautomerism with single-bond resolution. Single-molecule electronic techniques offer unlimited, unique opportunities to probe molecular mechanisms of chemical reactions and biological activities that are not accessible in conventional ensemble experiments.

## Methods

**Device fabrication and molecular connection**. The method for making the devices with graphene point contact arrays was described in Supplementary Figs. 1–3. In order to prevent any direct contact and leakage between the solution and metal electrodes during electrical measurements in TeCA or diphenyl ether, a 50 nm-thick silicon oxide layer ($SiO_2$, the transparent part on Au electrodes in Fig. 1a) was deposited by e-beam thermal evaporation after resistance thermal deposition of patterned metallic electrodes (8/60 nm, Cr/Au). After the DLL process, the freshly-prepared devices with a graphene point contact array were immersed in a pyridine solution containing 0.1 mM of the UPy-based molecule mentioned above and 1 mM EDCI (the coupling reagent). After 48 h, the devices were removed from the solution and then thoroughly rinsed several times with deionised water and $Me_2CO$, and dried with a $N_2$ stream. The as-prepared HBB-SMJ devices were ready for proving the successful connection under sequential water/diphenyl ether treatments.

**Electrical characterisation**. The preliminary electrical characterisation (I–V and I–t) was performed at room temperature in the ambient atmosphere by using an Agilent 4155 C semiconductor parameter system (DC measurements) and a Karl Suss (PM5) manual probe station. The real-time measurements of HBB-SMJs in TeCA and diphenyl ether were conducted by a ziControl program. The electrical signals from selected electrode pairs were collected by a trans-impedance current amplifier (HF2LI Lock-in Amplifier) sampled at 57.6 kHz with a NIDAQ card. The two-terminal device architecture was employed to characterise the device performance at the voltage bias of 300 mV.

**Temperature control**. Before real-time measurements, the devices were fixed on the testing stage, consisting of a manual probe station, an INSTEC hot/cold chuck (HCC214S, INSTEC), a proportion integration differentiation control system and a liquid $N_2$ cooling system. The temperature can be controlled with high resolution (0.001 °C) and ultrahigh temperature stability (better than ± 0.1 °C). We used a thermocouple to monitor the temperature of the solution. After the thermal equilibrium was reached at the objective temperature, we started to record the I–t signals of HBB-SMJs.

**Theoretical calculation**. Structure optimisations and electronic structure calculations were carried out within the Perdew–Burke–Ernzerhof generalised gradient approximation and implemented in Vienna ab initio simulation package[45]. The van der Waals (vdW) interaction was counted in by using the method of Grimme (DFT-D2)[46,47]. The solvent effect on the stability of various structures was also considered simply by adding a few solvent molecules around the structures. The projector augmented wave potential[48] and the plane-wave cut-off energy of 400 eV were used. A cubic supercell with the lattice constant of 20 Å and a Gamma k-point were employed. In the supercell, the positions of all atoms were relaxed until the force was less than 0.01 eV/Å. The criterion for the total energy was set as $1 \times 10^{-5}$ eV.

The reaction pathways for intermolecular proton transfer and lactam–lactim tautomerisation were calculated by the climbing image nudged elastic band method[49], which has been recognised to provide reliable reaction energy barriers and widely used in the literature. The quantum nature of the proton was not considered here, since in most cases, it is only important at low temperatures (usually smaller than 200 K). At room temperature, the proton transfer adopts the mechanism of thermally activated barrier crossing, and follows the Arrhenius equation[50–52]. We also note that a recent study reported that tunnelling played an important role in porphycene tautomerisation at room temperature[53], which will be taken into account in the further research.

The electronic transport simulations were performed by using real-space NEGF techniques implemented in the TranSIESTA package[54,55]. The PBE functional with vdW correction by using the method of Grimme (DFT-D2) was employed. A double zeta plus polarisation basis set and an energy shift of 0.01 Ry were used. The energy cutoff for real-space mesh size was set to be 200 Ry. The transport device model was divided into three parts, including left electrode, scattering region, and right electrode. Both electrodes were semi-infinite p-type doped graphene. The scattering region consisted of the molecule and one surface layer in both sides (Supplementary Fig. 13). The surface layer was constructed by one-unit cell of p-type doped graphene with partially oxidised edges.

**Data availability**. The data that support the findings of this study are available from the corresponding authors upon reasonable request.

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

## Acknowledgements

We are grateful to Y. Gao and L. Yang (Peking University, China) for discussions. We acknowledge primary financial support from National Key R&D Program of China (2017YFA0204901), the National Natural Science Foundation of China (grant 21373014, 21472196, 21521062 and 21688102), the National Key Research & Development Program of China (grant 2016YFA0200604), and the Strategic Priority Research Program of the Chinese Academy of Sciences (grant XDB 12010400). We used computational resources of Supercomputing Center of University of Science and Technology of China and Supercomputing Center of Chinese Academy of Sciences.

## Author contributions

X.G., J.Y., and Y.Z. conceived and designed the experiments. C.Z., J.C., Y.L., C.G. and G.H. performed the device fabrication and electrical measurements. Z.G. carried out the

molecular synthesis and $^1$H-NMR spectroscopy. X.L. performed the theoretical calculation. X.G., J.Y., Y.Z., C.Z., X.L. and Z.G. analysed the data and wrote the paper. All the authors discussed the results and commented on the manuscript.

## Additional information

**Competing interests:** The authors declare no competing financial interests.

