## [Peer Review File · Nature Communications]

Reviewers' comments:

Reviewer #1 (Remarks to the Author):

This is an excellent paper that convincingly demonstrates that tautomerization in H-bonded dimers can be monitored by electrical measurements on a single molecule (dimer in this case) level. I am impressed by the technical quality of the work. I would support publication, but only after introducing several corrections and extensions.

1. The first problem in this paper is, in my opinion, incomplete citation. In the introduction, the authors state: "...for the first time we represent direct real-time monitoring of single-molecule hydrogen bond dynamics on the basis of the platform of GM-SMJs...". While this whole statement is true, it may lead to a false impression that the observation of hydrogen bond dynamics has not been achieved so far for single molecules. In fact, double hydrogen transfer along two intramolecular hydrogen bonds has been observed for single molecules of porphycene using scanning probe microscopy and reported in several papers by Kumagai, Grill, and coworkers: (1) Kumagai et al., *Phys. Rev. Lett.* 2013, 111, 246101-246105; (2) Kumagai et al., *Nature Chem.* 2014, 6, 41-46; (3) J. N. Ladenthin et al., *ACS Nano* 2015, 9, 7287-7295; (4) H. Böckmann et al., *Nano Lett.* 2016, 16, 1034-1041; (5) J. Ladenthin et al., *Nature Chem.* 2016, 8, 935-940. Moreover, the methodology used in these works for rate determination (refs. (1) and (2) above) was practically the same as the approach used by the authors of the manuscript.

The authors quote a review paper on porphycene (ref. 35) that describes the results obtained by scanning probe microscopy, but the original works are not mentioned. This is an important point, since, without these quotations, the impression of novelty of the present work could be artificially enhanced.

2. With respect to the above, not only the appropriate papers should be quoted, but the merits and capabilities of the two techniques should be discussed.

3. While the technical part is brilliant, the discussion of the tautomerism mechanisms leaves some doubts. The authors decided to move most of the kinetic and computational data to the Supporting Information, which, I think, is justified, given that the stress is rather on the method, not on the details of the mechanisms of the observed phenomena. However, some important points are not clear. Figures S14 and S16 show the plots of energies vs "Reaction coordinate", without describing the latter. One does not learn how these energy profiles were calculated, and how reliable they are. It is well known that calculations of proton transfer are extremely challenging due to the quantum nature of the proton. Therefore, some comparison with previous studies would be advisable. For instance, do the calculated barriers agree with previous estimations?

4. On p.13 the authors state that the energy barrier of 0.67 eV can be overcome at room temperature. What is the basis for such a claim? kT at 293 K is only 0.026 eV.

5. I do not understand the difference between the terms "tautomerisation" and "proton transfer", used to indicate different equilibria. Proton transfer as well as tautomerization occur in each case.

6. Finally, in the studies of H-bonds, H/D substitution is usually one of the most instructive tools for studying structure and dynamics. It has not been mentioned in the manuscript. Perhaps the studies of isotopologues would be technically very difficult, but at least a discussion of such possibility should be included.

Reviewer #2 (Remarks to the Author):

Graphene nanogap devices are used here to examine a quadrupolar hydrogen bonding system which is covalently linked to both respective carbon electrodes through covalent amide linkages. At constant gap separation fluctuations in the current-time signal are observed which are assigned to hydrogen bond dynamics. Fluctuations are assigned to stochastic rearrangements of hydrogen bond structure through proton transfer and lactam-lactim tautomerism. This is clearly a complex molecular system with multiple tautomers and conformer possibilities. For studying hydrogen bond dynamics I would have preferred a simpler system exhibiting a single simple tautomerism or proton transfer possibility which would have streamlined the experimental and modeling possibilities thereby being more unequivocal.

Most importantly, before recommending publication, the authors need to provide clear and conclusive proof that the fluctuations do not arise from stochastic bond fluctuations which have been well documented in the literature. Here I recommend that they perform additional control experiments with molecular wires which does not have the hydrogen bond structure and proton transfer and tautomerism possibilities but possesses the same contacting to the graphene electrodes. Ideally this would be a molecular wire with similar conjugation and conductance values and the same amino end groups for linking with the carboxylic acid group terminated graphene point contacts. Note that the contacts used here may also have water or acid base sensitivity.

Listed below are the details of our point-by-point responses to the referees' comments.

Reviewer 1:

Comments: This is an excellent paper that convincingly demonstrates that tautomerization in H-bonded dimers can be monitored by electrical measurements on a single molecule (dimer in this case) level. I am impressed by the technical quality of the work. I would support publication, but only after introducing several corrections and extensions.

We thank the referee very much for his/her high evaluation and kind recommendation.

Comment 1: The first problem in this paper is, in my opinion, incomplete citation. In the introduction, the authors state: “..for the first time we represent direct real-time monitoring of single-molecule hydrogen bond dynamics on the basis of the platform of GM-SMJs...”. While this whole statement is true, it may lead to a false impression that the observation of hydrogen bond dynamics has not been achieved so far for single molecules. In fact, double hydrogen transfer along two intramolecular hydrogen bonds has been observed for single molecules of porphycene using scanning probe microscopy and reported in several papers by Kumagai, Grill, and coworkers: (1) Kumagai et al., Phys. Rev. Lett. 2013, 111, 246101-246105; (2) Kumagai et al., Nature Chem. 2014, 6, 41-46; (3) J. N. Ladenthin et al., ACS Nano 2015, 9, 7287-7295; (4) H. Böckmann et al., Nano Lett. 2016, 16, 1034-1041; (5) J. Ladenthin et al., Nature Chem. 2016, 8, 935–940. Moreover, the methodology used in these works for rate determination (refs. (1) and (2) above) was practically the same as the approach used by the authors of the manuscript.

The authors quote a review paper on porphycene (ref. 35) that describes the results obtained by scanning probe microscopy, but the original works are not mentioned. This is an important point, since, without these quotations, the impression of novelty of the present work could be artificially enhanced.

Answer: Thanks for the kind suggestion. We are sorry to omit these important studies. In response to this useful suggestion, we have added and cited these papers as Refs. 9-13 in the main text.

Comment 2: With respect to the above, not only the appropriate papers should be quoted, but the merits and capabilities of the two techniques should be discussed.

Answer: Thanks for the good suggestion. According to this suggestion, we further provided more discussion related to the merits and capabilities of the techniques. Please see Lines 8-19 in Page 3 of the main text.

Comment 3: While the technical part is brilliant, the discussion of the tautomerism mechanisms leaves some doubts. The authors decided to move most of the kinetic and computational data to the Supporting Information, which, I think, is justified, given that the stress is rather on the method, not on the details of the mechanisms of the observed phenomena. However, some important points are not clear. Figures S14 and S16 show the plots of energies vs “Reaction coordinate”, without describing the latter. One does not learn how these energy profiles were calculated, and how reliable they are. It is well known that calculations of proton transfer are extremely challenging due to the quantum nature of the proton. Therefore, some comparison with previous studies would be advisable. For instance, do the calculated barriers agree with previous estimations?

Answer: Thanks for the nice comments.

a) We are sorry for the missing information. The energy profiles in Figures S14 and S16 are calculated by the climbing image nudged elastic band (CI-NEB) method [*G. Henkelman and H. Jónsson, J. Chem. Phys. 113, 9901 (2000)*], which has been recognized to provide reliable reaction energy barriers and widely used in the literature. The CI-NEB method works by a three-step procedure: Firstly, a number of intermediate image structures are inserted along the reaction path between reactants and products; Secondly, all images are simultaneously optimized to find their lowest energy possible by minimizing only the component of the force perpendicular to the image band; Finally, the highest energy image is driven up to the saddle point. In the CI-NEB method, the reaction coordinate is just the location of each image, which can be indicated by either the distance from the reactant as in Figures S16 and S18 (original Figures S14 and S16), or the image number.

We have added more details in the captions of Figures S16 and S18 (original Figures S14 and S16), and also in the section of computational method (Please see Pages 18 and 19 in the main text, and Page S22 in the Supplementary Information).

b) The quantum nature of the proton is not considered in our calculations, since it is only important at low temperatures (usually smaller than 200 K). At room temperature, the proton transfer adopts the mechanism of thermally activated barrier crossing, and follows the Arrhenius equation [see *J. Phys.: Condens. Matter* 1, 9609 (1989); *J. Chem. Phys.* 89, 897 (1988); *J. Chem. Phys.* 95, 4201 (1991)]. A comment about this issue has been added in the revised manuscript (Please see Pages 18 and 19 in the main text, and Page S22 in the Supplementary Information).

c) Depending on the strength of hydrogen bond and its chemical environment, previous estimations of the barriers for intermolecular proton transfer vary from 1 kcal/mol to about 20 kcal/mol, i.e. 0.04 eV to 0.86 eV [see *J. Chem. Phys.* 79, 4694 (1983); *Acc. Chem. Res.* 18, 174 (1985); *J. Mol. Struct. (Theochem)* 307, 65 (1994)], while for lactam-lactim tautomerism, the energy barriers could change from 7 kcal/mol to 40 kcal/mol, i.e. 0.30 eV to 1.73 eV [see *PNAS* 110, 9243 (2013); *J. Chem. Theory Comput.* 5, 949 (2009)]. In our work, the calculated barriers for intermolecular proton transfer (0.35 eV) and lactam-lactim tautomerism (0.67 eV, 1.26 eV) fall reasonably in the range of previous estimations. Moreover, using the same methodology as ours, the energy barriers for proton-involved tautomerisation in melamine molecules have been predicted in very good agreement with the experiments [*PNAS* 106, 15259 (2009)]. A discussion about this issue has been added in the caption of Figure S16 (original Figure S14).

Comment 4: On p.13 the authors state that the energy barrier of 0.67 eV can be overcome at room temperature. What is the basis for such a claim? kT at 293 K is only 0.026 eV.

Answer: Thanks for the good questions. It is generally accepted that at room temperature, an energy barrier of approximately 0.75 eV corresponds to a reaction rate of 1 turnover per second (see *J. K. Nørskov, et al. Fundamental Concepts in Heterogeneous Catalysis, John Wiley & Sons, Inc: Hoboken, NJ, 2014, p9*). In particular, for our system, the reaction rate at room temperature

could be estimated according to the Arrhenius equation, $k = A\exp\{-E_a/k_B T\}$, where E_a is the energy barrier and A takes an experimental value of about 10^{13} s^{-1} [*PNAS* 110, 9243 (2013)]. This roughly gives a reaction rate of 60 turnover per second for an energy barrier of 0.67 eV. In this sense, we believe that the barrier can be overcome at room temperature. Moreover, the applied bias voltage potential in the experiment, and also the quantum tunneling effect of hydrogen (though small at room temperature), can both help overcome the barrier and increase the reaction rate.

Comment 5: I do not understand the difference between the terms “tautomerisation” and “proton transfer”, used to indicate different equilibria. Proton transfer as well as tautomerization occur in each case.

Answer: Sorry for this confusion. Strictly speaking, the lactam-lactim tautomerisation involves a proton transfer process. In fact, lactam-lactim tautomerisation is already a term commonly used in organic chemistry and biochemistry to describe a cyclic form of amide-imidic acid tautomerism in 2-pyridone and derived structures. To keep consistent, we would like to remain this term in the manuscript. At the same time, to avoid any confusion, we changed ‘proton transfer’ to ‘intermolecular proton transfer’, since lactam-lactim tautomerisation is intramolecular. Thanks for the good suggestion.

Comment 6: Finally, in the studies of H-bonds, H/D substitution is usually one of the most instructive tools for studying structure and dynamics. It has not been mentioned in the manuscript. Perhaps the studies of isotopologues would be technically very difficult, but at least a discussion of such possibility should be included.

Answer: Thanks for the good comment. We added this part (Page 3) and cited the related papers as Refs. 3-4.

Reviewer 2:

Comments: Graphene nanogap devices are used here to examine a quadrupolar hydrogen bonding system which is covalently linked to both respective carbon electrodes through covalent amide linkages. At constant gap separation fluctuations in the current-time signal are observed which are assigned to hydrogen bond dynamics. Fluctuations are assigned to stochastic rearrangements of hydrogen bond structure through proton transfer and lactam-lactim tautomerism.

Thank the referee very much for his/her high evaluation and kind recommendation.

Comment 1: This is clearly a complex molecular system with multiple tautomers and conformer possibilities. For studying hydrogen bond dynamics I would have preferred a simpler system exhibiting a single simple tautomerism or proton transfer possibility which would have streamlined the experimental and modeling possibilities thereby being more unequivocal.

Answer: Thanks for the useful suggestion. We agree with the comment that a simpler system is a good choice. Considering the symmetry, double hydrogen-bond and quadruple hydrogen-bond supramolecular systems meet the requirements because the same monomer need to be linked on the edges of graphene point contacts. In response to this kind advice, we have tried a double hydrogen-bond system exhibiting a single proton transfer possibility (Supplementary Figure 10a). Even though similar on-off switching behaviors by different solvent treatments (Supplementary Figure 10b) and two-level fluctuations in $I-t$ measurements (Supplementary Figures 10c and 10d) can be observed, the yield of the device fabrication is lower (~2%) and the devices showed the weak endurance of long-time measurements and higher temperatures. The corresponding theoretical calculations illustrated the energy of the intermolecular proton transfer process in the double hydrogen-bond system (Supplementary Figures 10e and 10f).

Comment 2: Most importantly, before recommending publication, the authors need to provide clear and conclusive proof that the fluctuations do not arise from stochastic bond fluctuations

which have been well documented in the literature. Here I recommend that they perform additional control experiments with molecular wires which does not have the hydrogen bond structure and proton transfer and tautomerism possibilities but possesses the same contacting to the graphene electrodes. Ideally this would be a molecular wire with similar conjugation and conductance values and the same amino end groups for linking with the carboxylic acid group terminated graphene point contacts. Note that the contacts used here may also have water or acid base sensitivity.

Answer: Thanks for the comment and helpful suggestion. To prove the fact that the fluctuations only originate from the stochastic rearrangement of the hydrogen-bond structure, we designed and synthesized a control molecule based on a sexiphenyl structure, which has similar conjugation and conductance values, and the same amino end groups with the quadruple hydrogen-bond supramolecular system. On the basis of the investigation of the current-time ($I-t$) characteristics of single-molecule junctions reconnected by this control molecule under the same measurement condition, we did observe no particular fluctuations (Supplementary Figure 9). This is consistent with the explanation that the observed stochastic switching fluctuations in HBB-SMJs should originate from the hydrogen-bond structure rather than stochastic bond fluctuations. We added the corresponding discussion into the revised manuscript on Page 8 in the main text.

Finally, we would like to thank all the referees very much for the helpful suggestions, the patience, the time, and the kind recommendations.

REVIEWERS' COMMENTS:

Reviewer #1 (Remarks to the Author):

I can now recommend this work for publication. The authors have addressed my comments. Still, the reply b) to comment 3 leaves some doubts. The statement that the quantum nature of the proton is only important at low temperatures may not always BE true. This has been recently demonstrated for double hydrogen transfer in porphycene (J. Phys. Chem. Lett., 2016, 7, 283–288). I suggest that this finding be added to the discussion on p.19.

Minor language comments:

- p.3: "unitized" >> utilized?
- p. 13: "In another word" >> "In other words"
- ref. 52: "yemperature"

Reviewer #2 (Remarks to the Author):

I have examined the changes made Professor Guo and co-workers on resubmission of the manuscript entitled "Direct observation of single-molecule hydrogen-bond dynamics with single-bond resolution". The authors have done a very thorough reply and modifications and additional control experiments. I am very happy with the changes and the modification and I am happy to recommend publication in this revised form. Most important for me was the control experiments on the sexiphenyl structure which does not show the current fluctuations which shows that the fluctuations seen for the lactam-lactim junctions can be assigned to stochastic rearrangements of hydrogen bond structure through proton transfer and lactam-lactim tautomerizism.

Listed below are the details of our point-by-point responses to the referees' comments.

Reviewer #1:

Comments: I can now recommend this work for publication.

We thank the referee very much for his/her high evaluation and kind recommendation.

Comment 1: The authors have addressed my comments. Still, the reply b) to comment 3 leaves some doubts. The statement that the quantum nature of the proton is only important at low temperatures may not always BE true. This has been recently demonstrated for double hydrogen transfer in porphycene (J. Phys. Chem. Lett., 2016, 7, 283–288). I suggest that this finding be added to the discussion on p.19.

Answer: Thanks for the good suggestion. We are sorry to omit this important finding. According to this suggestion, we added the discussion and cited this paper as Ref. 53. Please see Lines 405-408 in Page 19 of the main text and Ref. 53 cited in Pages 23-24 of the main text.

Comment 2:Minor language comments:

- p.3: “unitized” >> utilized?
- p. 13: “In another word” >> “In other words”
- ref. 52: “yemperature”

Answer: Thanks for the nice suggestions. We have corrected these spelling mistakes in Line 69 in Page 3, Line 276 in Page 13 and Line 570 in Page 23.

Reviewer #2 (Remarks to the Author):

I have examined the changes made Professor Guo and co-workers on resubmission of the manuscript entitled "Direct observation of single-molecule hydrogen-bond dynamics with single-bond resolution". The authors have done a very thorough reply and modifications and additional control experiments. I am very happy with the changes and the modification and I am happy to recommend publication in this revised form. Most important for me was the control experiments on the sexiphenyl structure which does not show the current fluctuations which shows that the fluctuations seen for the lactam-lactim junctions can be assigned to stochastic rearrangements of hydrogen bond structure through proton transfer and lactam-lactim tautomerizism.

We thank the referee very much for his/her high evaluation and kind recommendation.

Finally, we would like to thank all the referees very much for the helpful suggestions, the patience, the time, and the kind recommendations.